# Structural Optimization of Lightweight Composite Floors with Integrated Constrained Layer Damping for Vibration Control

Carlos M. C. Renedo [1], Iván M. Díaz [1,*], Jaime H. García-Palacios [2] and Christian Gallegos-Calderón [1]

[1] Department of Continuum Mechanics and Theory of Structures, ETSI Caminos, Canales y Puertos, Universidad Politécnica de Madrid, Calle Profesor Aranguren 3, 28040 Madrid, Spain; carlos.martindelaconcha@upm.es (C.M.C.R.); christian.gallegos@upm.es (C.G.-C.)

[2] Department of Hydraulics, Energy and Environmental Engineering, ETSI Caminos, Canales y Puertos, Universidad Politécnica de Madrid, Calle Profesor Aranguren 3, 28040 Madrid, Spain; jaime.garcia.palacios@upm.es

* Correspondence: ivan.munoz@upm.es

**Abstract:** Due to current architectural trends, contemporary public buildings are becoming open-plan spaces with much less weight and damping. Consequently, Vibration Serviceability Limit State (VSLS) due to human-induced vibrations has become an increasing concern for structural engineers, especially when designing offices, hospitals, or gymnasiums. When dealing with resonant vibrations, a slight increase in the floor-damping enables decreasing considerably the vibration level. The damping strategy studied in this work is usually known in the literature as Constrained Layer Damping (CLD) and consists of a viscoelastic layer constrained between the concrete slab and the steel beam of a lightweight composite floor. In this paper, a complete structural checking methodology has been developed for analyzing all the limit states that determine the final sizing of a steel–concrete composite floor treated with CLD, including a detailed analysis of the VSLS. The methodology has been used for setting a structural optimization problem for floors with and without CLD treatments. Thus, it has been demonstrated that the integration of CLD treatments at the design stage of the building allows the development of lighter floor structures with a smaller embodied carbon (EC) footprint, especially for long-span schemes with restrictive vibration limitations.

**Keywords:** floor vibration; viscoelastic damping; structural optimization; vibration control

## 1. Introduction

Currently, composite floors are widely used in multistory commercial and office buildings due to their capability to span longer distances with stronger, stiffer, and lighter structural solutions, which are economical and offer a fast construction process. Their use often implies the reduction of the primary structure and the foundations of the building, thus reducing its overall embodied carbon (EC) [1]. Over recent decades, the increasing use of large open-plan spaces and the rise of the electronic office has led to the reduction of dead and live loads (1 kPa less) [2,3] due to the removal of full-height partitions and heavy furnishing elements. As a consequence, modern composite floors have lower damping ratios (around 1% to 3% [4–7]), lower mass, and lower fundamental natural frequencies (often below 10 Hz), making them more prone to vibrate under human-induced dynamic loading, such as walking in offices or rhythmic activities in gymnasiums [8]. Furthermore, restrictive vibration comfort levels (0.02–0.04 m/s$^2$) are presently required for floors in calm environments such as residences, offices, and hospitals. In this context, the Vibration Serviceability Limit State (VSLS) has become a sizing criterion to consider when designing composite floors [9].

Excessive vibrations are often related to low-frequency floors (LFF) (with natural frequencies below 10 Hz) due to their resonance with the 2nd, 3rd, or even 4th harmonic of the human dynamic loading. In high-frequency floors (HFF) (above 10 Hz) it is considered that

the vibration does not have a resonant nature and is made of consecutive transient decaying responses of the floor to each human footfall [5]. Knowing this, structural designers often try to include stiffer-enough steel members (that raise the natural frequencies of the floor to avoid resonance) or heavy concrete slabs that decrease the amplitude of the vibration [10]. This practice leads to a significant increase in the material used on the floor (especially relevant when the vibration limits to comply are very restrictive) affecting its lightness and increasing its EC as has been proved by Gonçalves and Pavic in [9]. This research studies the structural oversizing to be performed in a composite floor of 30 m × 44 m with bays of 6 m × 10 m when the VSLS to be met becomes more restrictive (changing from a limiting root mean square (rms) acceleration of 0.04 m/s$^2$ to 0.02 m/s$^2$). They concluded that the floor weight should be increased by 27% and its EC by 14% to accomplish the new comfort level. Alternatively, the integration of damping strategies into the floor's design may be an effective way of improving their dynamic performance without increasing their structural mass, thus, minimizing their EC. This is a key point considering that floors are responsible for around 70% of the total EC of the superstructure of a building [11,12].

Two main types of damping strategies may be used to solve floor vibration problems. First, the use of inertial dampers, in their passive versions (most known as Tuned Mass Dampers or TMDs) [13], semi-active [14] and active versions [15] that apply counteract forces in real time to reduce the structural motion. Second, the use of dissipative strategies that enable increasing the inherent damping of the floor by wisely including specific elements that dissipate additional energy, such as viscous or Viscoelastic (VE) dampers. This paper studies one damping treatment related to this second group: Constrained Layer Damping (CLD). It consists of a thin VE layer embedded between two elastic bending members. The VE layer suffers shear deformation when the constraining elastic members vibrate in bending modes, hence additional energy is dissipated through a stress-strain shear hysteresis [16].

CLD treatments were developed and have been extensively applied in aerospace and mechanical engineering to mitigate broad-band frequency vibrations [17]. Their use in civil engineering is still limited, although, some studies have been carried out using them to improve the dynamic performance of composite floors. First, Nelson [18] (Figure 1a) and Farah et al. [19] (Figure 1b) applied a VE CLD treatment to the lower steel flanges of a lively floor, obtaining damping ratio increases from 3.5% to 5%. Ebrahimpour [20] (Figure 1c) confirmed these results with a similar treatment. Later, Ahmadi et al. [21] (Figure 1d) developed a new CLD retrofitting system applied to the concrete slab of the floor with similar damping ratio increases. In 2006, ARUP in collaboration with Richard Lee Steel Decking proposed the first commercial CLD solution to be integrated into a composite floor since the construction stage. This was called 'Resotec' (Figure 1e), and its applicability was described by Willford et al. [22] (Figure 2). This type of CLD treatment is the one studied in this paper. The major advance of this proposal over the previous ones was the location of the VE layer between the rib-deck concrete slab and the upper flange of the steel beam (as close as possible to the composite section centroid) where the shear strain to be achieved is maximum. Moreover, they recommended including the VE layer only for a percentage of the beam's length near the supports, where the longitudinal shear strain is higher, whereas the central part of the composite beam remains connected to the shear using studs. Therefore, there is a trade-off between the achieved inherent damping and the loss of stiffness due to the partial shear connection in length. The CLD treatment itself used by ARUP has an overall thickness of 3 mm and is composed of two thin steel sheets of 1 mm that constrain a slim VE layer of 1 mm. Unlike previous applications, this CLD treatment was not conceived to be retrofitting solution but an integrated technology that needed to be considered when designing the composite floor. The optimal design of this CLD treatment and its integration into the design workflow of a composite floor were aspects not covered by Willford et al. [22].

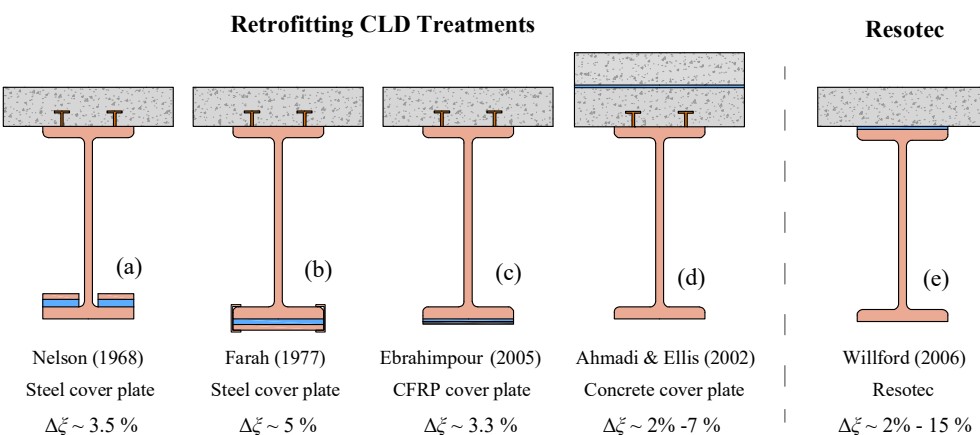

**Figure 1.** Evolution in the application of CLD treatments to composite floors. (**a**) CLD Application of Nelson [18] over the lower flange. (**b**) CLD Application of Farah et al. [19] under the lower flange. (**c**) CLD Application of Ebrahimpour [20] with a CFRP constraining layer. (**d**) CLD Application of Ahmadi et al. [21] with a constraining concrete slab. (**e**) Resotec CLD Application developed by Willford et al. [22]

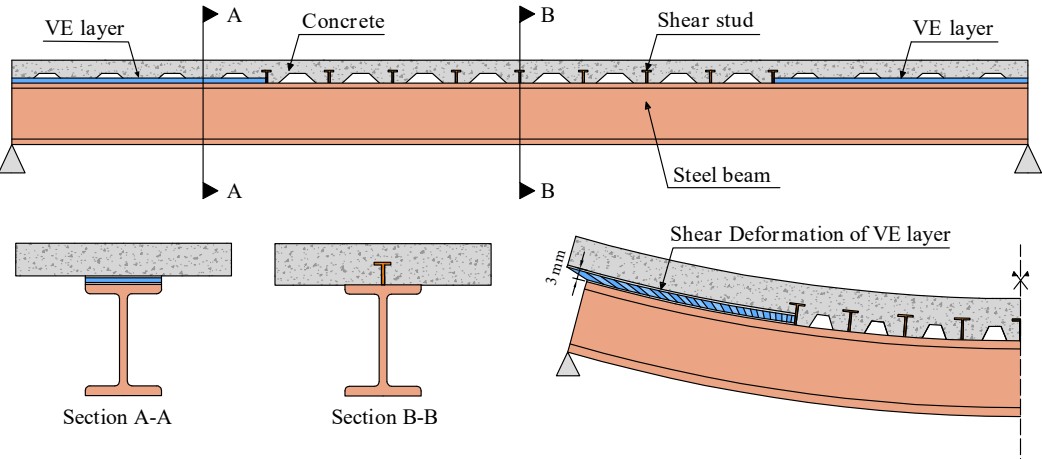

**Figure 2.** '*Resotec*' CLD treatment installed along 50% of the beam's length (50% CLD) Willford et al. [22].

Thus, this paper aims to provide a methodology for the structural checking and structural optimization of composite floors prone to vibrate with this integrated CLD treatment. The main objective is to prove that the use of this CLD treatment enables overcoming the VSLS without adding great amounts of additional mass to the floor and minimizing its EC.

The structural optimization of composite floors has been researched by many authors using different optimization algorithms, Objective Functions (OFs), and problem constraints (which are the limits states (LSs) to be met by the floor). Initially, the authors used the weight and the cost of the floor as OFs. Zahn [23], optimized the weight of a set of composite beams using two different codes. The VSLS was considered according to an early recommendation given by Murray in 1981 [24] based on a heel drop loading. He concluded that vibrations were the sizing criterion for spans between 3 m and 6 m. Later, Kim and Adeli [25,26] optimized the economic cost of composite floors with a simple OF considering the influences of concrete, steel, and studs. They affirmed that the VSLS was checked but without providing details. The cost optimization performed by Klanšek and Kravanja [27–29] used a complex OF accounting for material, energy consumption, and labor cost items required to manufacture a composite floor. They studied the use of different

types of steel members for typical ranges of spans and loads, according to the Eurocodes but without considering the VSLS. The research developed by Kaveh and Ahangaran [30,31] followed a similar methodology as the one proposed by Zahn but they focused on testing the efficiency of novel meta-heuristic algorithms for structural optimization. Poitras et al. [32] was the first author to include a widely accepted methodology to check the VSLS (the AISC Steel Design Guide 11 Floor vibrations due to human activity [7]) for the optimization process of two composite floor examples. His work was used as a reference for another research performed by Kaveh and Ghafari [33] where it was concluded that the vibrations were one of the critical LSs when designing a floor bay of 8 m × 10 m. Yossef and Taher [34] performed another cost optimization of composite floors with castellated beams using a widely accepted methodology to check the VSLS (the SCI guideline Design of floors for vibration [5]). In recent years, with the rise of life-cycle assessment, many authors have started to include the EC (measured in $kgCO_2eq$) of the floor as an OF to optimize. Roynon [35] has presented a manual optimization of framed buildings minimizing their EC and giving reference values of $kgCO_2eq/m^2$ for steel framed floors (between 80–250 $kgCO_2eq/m^2$ for regular spans and 150–300 $kgCO_2eq/m^2$ for long-span floors). Drevniok et al. [36] have developed an optimization methodology called 'The Lightest Beam Method' to study the amount of material that could have been reduced in a set of already designed and built steel floors without composite action. They consider the VSLS as a limitation in the natural frequency of 3 Hz, without assessing in detail the level of vibration. They concluded that the relaxation of serviceability requirements at the design stage could enable a reduction by more than 30% of the floors' weight. Some authors as Whitworth and Tsavdaridis [37] and Kravanja et al. [38] have used the embodied energy (measured in J) to quantify the environmental impact of the floors. Finally, in the research performed by Gauch et al. [39] different structural types of floors (made of concrete, steel, and timber) are designed and compared in terms of cost and EC obtaining interesting conclusions especially applicable to a decision-making stage of the design process. They again consider the VSLS by imposing a limit on the floor fundamental frequency of 4 Hz.

As a conclusion, it is clear that the VSLS should be properly taken into account in terms of the floor's acceleration response according to currently accepted guidelines as [5,7] or [6]. Thus, to the author's knowledge, this paper is the first one addressing the multi-objective optimization (with the floor's Weight and its EC as Ofs) of composite floors designed with and without integrated CLD treatments, and including a detailed verification of the VSLS.

The remainder of this paper is organized as follows: Section 2 describes the elements of the type of composite floors to be optimized. Section 3 outlines all the static limit states to be checked along the floor design. Section 4 explains the workflow used to check the VSLS and how the CLD treatment has been optimally designed for each analyzed case. Section 5 portrays the multi-objective structural optimization carried out describing the optimization procedure, the Ofs, and the optimization parameters used. Section 6 describes a parametric study in which different square floor bays with different spans have been optimized with and without the use of CLD treatments to illustrate the proposed methodology. Finally, Section 7 provides some conclusions.

## 2. Description of a Composite Floor with Integrated CLD Treatment

A composite floor with a given total length $L$ and a total width $B$ can be composed of different bays as represented in Figure 3. A bay is defined as a floor part between 4 columns and is composed of primary ($1^{ry}$) beams spanning between columns and secondary ($2^{ry}$) beams that span between these first ones. There are 3 types of bays depending on their location, (i) corner bays, (ii) edge bays, and (iii) internal bays. The first two groups are usually more prone to vibrate as they cannot mobilize as much mass when vibrating as the internal bays.

In this paper, the design of a composite floor with integrated CLD treatments is carried out according to the floor bay depicted in Figure 4. A given floor bay is geometrically defined by the length of its $1^{ry}$ and $2^{ry}$ beams, here known as $L_1$ and $L_2$, respectively; the

number of $2^{ry}$ beams $N_2$ and the distance between them $d_2$. The steel members of the floor are here defined through profile sections from the Universal Beam (UB) series (this catalog was chosen as it is large enough so that a meaningful optimization can be carried out). The profiles are represented with the integer values $P_1$ and $P_2$ which correspond with the profile positions in the UB catalog sorted according to the moment of inertia. S-275 steel has been considered. The concrete slab of the floor is defined by two parameters, (i) the commercial rib-deck used for the floor, which is defined with the integer value $Rd$ (according to its position on the list of Cofraplus 60 rib-decks, with 4 different gauge thicknesses of 0.7 mm; 0.9 mm; 1 mm and 1.2 mm), and, (ii) the concrete slab thickness over the ribs in centimeters represented with the integer value $hc$. The concrete grade used is C-30. Steel studs have been included along the central region of the $1^{ry}$ and $2^{ry}$ beams to ensure a certain degree of shear connection between the steel members and the concrete slab. In the $1^{ry}$ beams, the separation of studs depends on the required bending resistance of the composite section at mid-span. In the $2^{ry}$ beams their separation is equal to the distance between the valleys of the rib-deck. Ductile shear studs made of steel with a diameter of 19 mm, an as-welded height of 95 mm, and an ultimate limit stress of $f_u$ = 450 Mpa, have been considered.

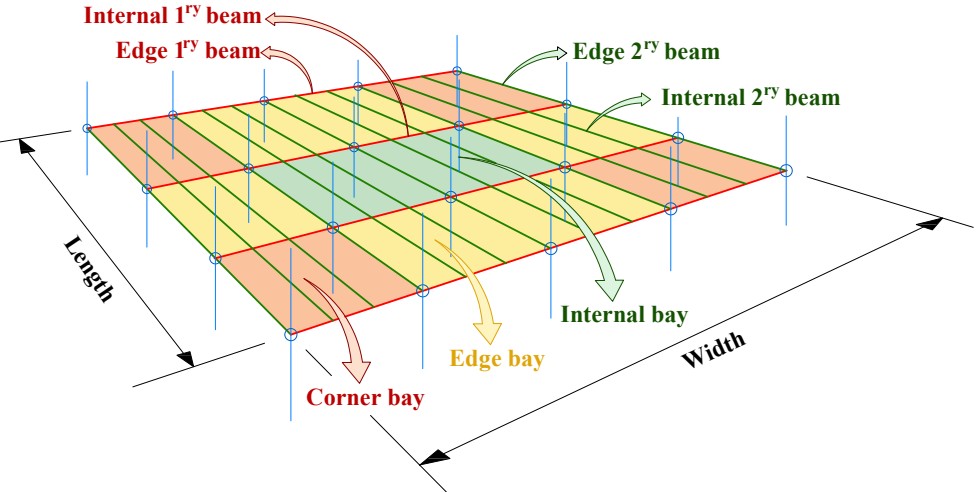

**Figure 3.** Scheme of a floor layout composed of bays.

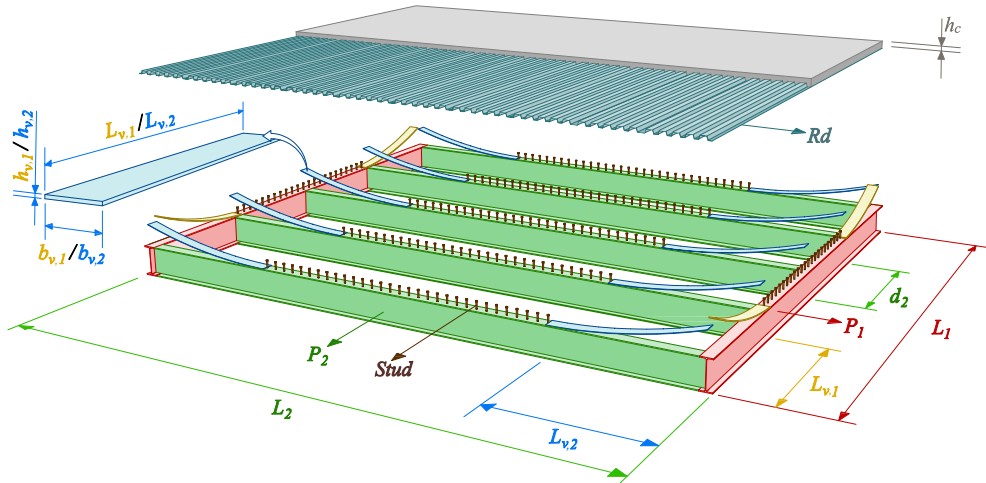

**Figure 4.** Elements of a floor bay with an integrated CLD treatment.

The CLD treatment has been included in all beams along a given length, $L_{v,i}$, near the supports (being "$i$" an index used to refer the $1^{ry}$ or $2^{ry}$ beam). Thus, the percentages of beam" length treated with CLD are obtained as $\%CLD_i = 100 \, \lambda_i$, where $\lambda_i$ is obtained as follows:

$$\lambda_i = (2\,L_{v,i})/L_i. \tag{1}$$

The thickness of the VE layer of these CLD treatments, $h_{v,i}$, needs to be small so three possible values of 0.5 mm; 1.0 mm and 2 mm have been considered for this parameter based on [40]. Moreover, the width of these VE layers, $b_{v,i}$, should be similar to the width of the top flange of their correspondent beams, $b_{f,i}$. For that reason, $b_{v,i}$ must take a value within the continuous range that goes from 0.5 $b_{f,i}$ to 2 $b_{f,i}$. The properties assigned to the VE material of the CLD treatment are in the range of commercial VE materials for vibration damping (for example, the HIP2, before cited [40]) i.e., a storage shear modulus $''_v$ of 0.5 mPa, a loss factor $\eta_v$ with a value of 1 and a density of 1700 kg/m$^3$. The loss factor of this type of material is usually measured based on the standard testing method ASTM:E756-05 [41].

In Table 1 the main material properties used along the paper are listed indicating, the name of the material, the element in which it is used, and its main mechanical properties. $\rho$ is used to denote the density of the material, $E_{st}$ for the Young modulus of the steel, $E_c$ for the Young modulus of the concrete, $f_y$ is the yielding strength of the steel, and $f_u$ is the ultimate tensile strength of the steel.

**Table 1.** Material properties of the floor.

| Material | Element | $\rho$ [kg/m$^3$] | $E_{st}$ [gPa] | $f_y$ [mPa] |
|---|---|---|---|---|
| Steel S275 | Profiles | 7850 | 210 | 275 |
| Steel B500S | Reinforcement | 7850 | 210 | 500 |
| Steel S350GD | Rib-deck | 7850 | 210 | 350 |
| | | $\rho$ [kg/m$^3$] | $E_{st}$ [gPa] | $f_u$ [mPa] |
| Steel S235J2+ | Shear studs | 7850 | 190 | 450 |
| | | $\rho$ [kg/m$^3$] | $E_c$ [gPa] | $f_{ck}$ [mPa] |
| Concrete C30 | Slab | 2500 | 30 | 30 |
| | | $\rho$ [kg/m$^3$] | $''_v$ [mPa] | $\eta_v$ [-] |
| HIP2 | CLD treatment | 1700 | 0.7 | 1 |

In this paper, for given a floor bay defined by a set of $L_1$ and $L_2$, the rest of the floor parameters above described ($P_1$, $P_2$, $N_2$, $h_c$, $R_d$, $L_{v,1}$, $L_{v,2}$, $h_{v,1}$, $h_{v,2}$, $b_{v,1}$ and $b_{v,2}$) are optimized to minimize two oFs, while meeting all the LSs of the floor design.

## 3. Static LSs to Be Met by the Floor

### 3.1. Static LSs in the 1$^{ry}$ and 2$^{ry}$ Beams

All the beams of the floor are designed as simply supported and built without any propping system. Hence, two design stages have been verified: (i) the construction and (ii) the service life of the floor. The LSs checked in the 1$^{ry}$ and 2$^{ry}$ beams are summarized in Table 2. Each LS is defined through a safety factor $SF$, which, if higher than 1, means that LS is met. To identify each $SF$, a set of three subscripts separated by commas has been used:

- *Subscript 1* identifies the variable used to compute the safety factor. The following options are available: '$M+$' means the sagging bending moment at mid-span, '$M+B$' means the sagging bending moment at the location of the first shear stud in partially treated beams, '$V$' denotes shear force at the support, and '$\delta LL$' means deflection under live load action.
- *Subscript 2* may be '$c$' or '$s$' depending on if the LS belongs to the construction stage or to the service-life stage, respectively.
- *Subscript 3* is the subscript '$i$' used to denote the 1$^{ry}$ or 2$^{ry}$ beam.

**Table 2.** Static LSs checked in the $1^{ry}$ and $2^{ry}$ beams of the floor with integrated CLD treatment.

| Stage | Limit State | Type | Location | Loads | Section | Safety Factor |
|---|---|---|---|---|---|---|
| Construction | ULS | Bending | *(simply supported beam, load at mid-span)* | $SW + LL_c$ | Steel | $SF_{M+,c,i}$ |
| | | Shear | *(simply supported beam, load at support)* | $SW + LL_c$ | Steel | $SF_{V,c,i}$ |
| Service-Life | ULS | Bending | *(composite beam with distributed load, point near end of CLD)* | $SW + DL + LL_s$ | Steel | $SF_{M+B,s,i}$ |
| | | Bending | *(composite beam with distributed load, load at mid-span)* | $SW + DL + LL_s$ | Composite | $SF_{M+,s,i}$ |
| | | Shear | *(composite beam with distributed load, load at support)* | $SW + DL + LL_s$ | Steel | $SF_{V,s,i}$ |
| | SLS | Deflection | *(composite beam with distributed load, load at mid-span)* | $LL_s$ | Composite | $SF_{\delta LL,s,i}$ |

### 3.1.1. Static LSs of $1^{ry}$ and $2^{ry}$ Beams: The Construction

The loads considered here are the self-weight, $SW_c$ and a construction live load $LL_c$ = 0.75 kN/m$^2$. At this stage, the concrete is not a resisting element and it has a fresh density of 2500 kg/m$^3$.

The verification has been done according to Eurocode 3 [42], as the only resisting element is the steel member of the floor. The Ultimate LS (ULS) of bending at mid-span has been checked with respect to the plastic bending moment of the steel section, $M_{p,st}$, and the ULS of shear at the support section with respect to the plastic shear of the steel web, $V_{p,st}$ (with the subscript '$p$' indicating plastic and '$st$' indicating steel).

### 3.1.2. Static LSs of $1^{ry}$ and $2^{ry}$ Beams: The Service Life

The loads considered at this stage are the self-weight $SW_s$ of the structural elements, (but now using a dried density for the concrete of 2400 kg/m$^3$), a deal load $DL_s$ for flooring finishes of 1 kN/m$^2$ and a live load $LL_s$ of 3 kN/m$^2$.

The verification has been performed following Eurocode 4 [43], as the concrete slab is now a resisting element of the beam section. Four LSs have been checked at this stage, (i) the ULS of bending at the end of the CLD treatment, where the fist shear stud is located (ii) the ULS of bending at mid-span, (iii) the ULS of shear, verified only considering the resisting contribution of the web of the steel member, and (iv) the deflection serviceability LS (DSLS) at mid-span under the action of $LL_s$, which has been limited to $L_i/350$. Special attention must be paid to the verification of the ULS of bending and the DSLS of composite beams partially treated with CLD along their length.

### 3.1.3. ULS of Bending for Composite Beams Partially Treated with CLD

In a regular simply supported composite beam, the ULS of bending is verified by assuring that the resisting moment of the composite section at mid-span $M_{Rd,comp}$ is higher than the service life acting design bending moment $M_{Ed,s}$. This criterion is not sufficient

for partially treated beams as the most critical section for bending may lie somewhere in between the first section connected to the shear and the mid-span section. This can be seen in Figure 5, where the bending resisting envelopes of a 15 m secondary beam treated with different %*CLD* have been compared with the laws of maximum design bending moment. It can be noted that along the CLD-treated length, the resisting bending moment is the one provided by the steel member, $M_{Rd,st}$, whereas, along the shear-connected central length its value increases due to the composite action. A simplified methodology to avoid checking the ULS of bending along the whole beam length has been developed based on checking two critical sections, (i) the mid-span and (ii) the first shear-connected section, here called Section B. A similar simplification was performed by Willford et al. [22]; However, in this contribution, it was assumed a degree of shear connection along the central connected region of 60% and it did not provide a detailed explanation of how this ULS should be computed. A more accurate description of this verification is provided below with adequate assumptions.

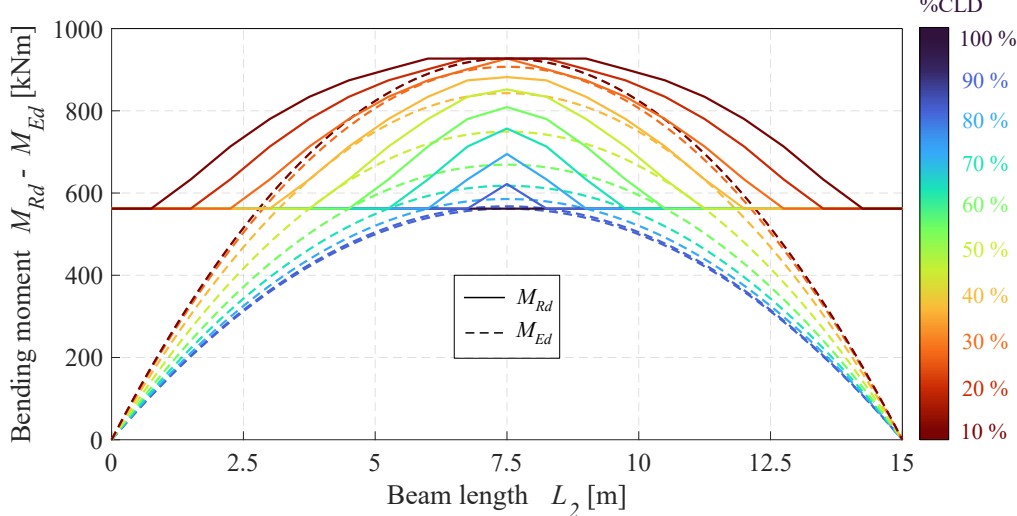

**Figure 5.** Envelopes of resisting bending moment, $M_{Rd}$, compared to the laws of maximum design bending moment, $M_{Ed}$, for different %*CLD* treatment.

For checking the Section B the design bending moment at this section has been compared to the plastic bending moment of the steel member, $M_{p,st}$.

To verify the mid-span section the following considerations have been made:

- The effective breadth of the slab, $b_{eff,i}$ , contributing to the composite section has been computed as follows:

$$b_{eff,1} = \min \left\{ \begin{array}{c} (L_1 - 2L_{v1})/4 \\ L_2 \end{array} \right\}, \tag{2}$$

$$b_{eff,2} = \min \left\{ \begin{array}{c} (L_2 - 2L_{v2})/4 \\ d_2 \end{array} \right\}. \tag{3}$$

The expression used here is the same one as for regular simply supported composite beams, but replacing the beam length by the shear-connected length, $L_{conn,i} = (L_i - 2\,L_{v,i})$. This decision is proposed by the authors as it is on the safe side.

- Two ductile shear studs per section connected have been used to achieve a higher degree of shear connection in a shorter distance. Additionally, plastic redistribution of longitudinal shear forces between the different shear studs along the connected length has been assumed.

- Partial shear connection theory has been assumed. Hence, the maximum longitudinal shear to be transferred within the steel–concrete interface, $N_c$ is limited by the plastic capacity of the shear connection. This means that the maximum longitudinal shear

when considering full shear connection, named $N_{cf}$, needs to be reduced by a factor $\alpha$ called 'degree of shear connection' which is computed as follows:

$$\alpha = \frac{P_{rd}(n/2)}{N_{cf}}, \tag{4}$$

$$N_c = \alpha N_{cf}, \tag{5}$$

where $P_{rd}$ is the shear resisting value of a single shear stud and $n$ is the total number of shear studs along the shear-connected length of the beam.

Finally, the resisting bending moment at mid-span has been computed following the methodology given in Eurocode 4 [43].

### 3.1.4. DSLS for Composite Beams Partially Treated with CLD

The authors propose the use of two bending stiffnesses along the beam length (see Figure 6): (i) the stiffness of the steel member for the CLD-treated region, and (ii) the bending stiffness of the composite section considering full-composite action (i.e., an infinitely rigid shear connection) and the cracking of the concrete slab. The slab" effective breadth used has been the same as the one described for the ULS given in Equations (2) and (3).

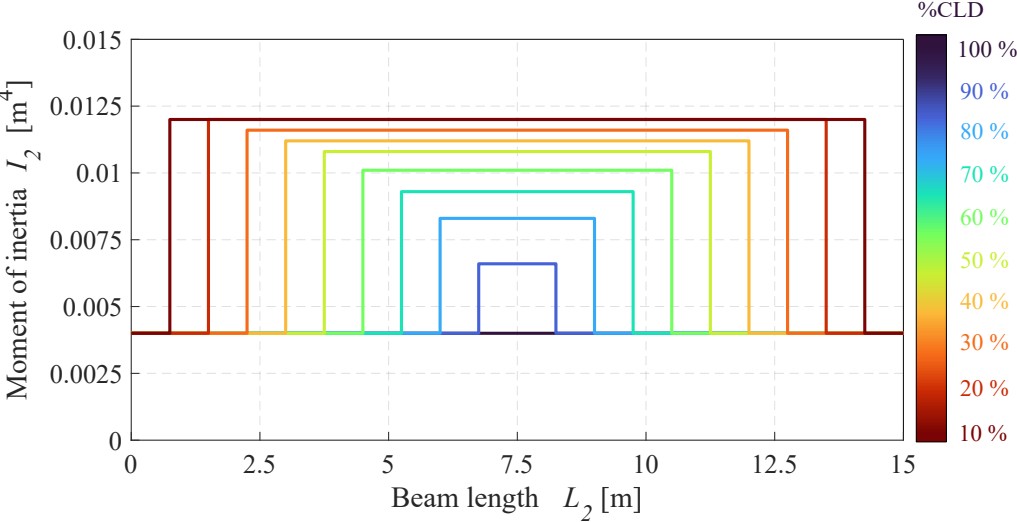

**Figure 6.** Equivalent moments of inertia homogenized to the concrete used to compute the deflection of beams with different *%CLD* treatment.

### 3.2. Static LSs in the Rib-Deck Slab

The rib-deck slab has been analyzed as a multi-span beam with as many supports as secondary beams are used within a floor bay. Again, two stages have been checked: (i) the construction and, (ii) the service life of the rib-deck. Table 3 summarizes the set of LSs checked in the rib-deck slab.

Again, each LS is related to a *SF* defined with a set of two subscripts:

- Subscript 1 defines the variable used to compute the *SF*: '*M*+' refers to the sagging bending moment at the edge span of the slab, '*M*−' denotes the hogging bending moment at the internal support of the edge span, '*V*' means the shear force at the internal support of the edge span, '*δLL*' means the maximum deflection under the live load action at the edge span, and '*δSW*' indicates the maximum deflection under the self-weight action at the edge span.
- Subscript 2 again may be '*c*' or '*s*' depending on the stage analyzed.

**Table 3.** Static LSs checked in the rib-deck slab of the floor.

| Stage | Limit State | Type | Location | Loads | Section | Safety Factor |
|---|---|---|---|---|---|---|
| Construction | ULS | Bending | | $SW + LL_c$ | Steel | $SF_{M+,c}$ $SF_{M-,c}$ |
| | | Shear | | $SW + LL_c$ | Steel | $SF_{V,c}$ |
| | SLS | Deflection ponding | | $SW + LL_c$ | Steel | $SF_{\delta LL,c}$ |
| Service-Life | ULS | Bending | | $SW + DL + LL_s$ | Composite | $SF_{M+,s}$ $SF_{M-,s}$ |
| | | Shear | | $SW + DL + LL_s$ | Composite | $SF_{V,s}$ |
| | SLS | Deflection | | $LL_s$ | Composite | $SF_{\delta LL,s}$ |

### 3.2.1. Static LSs in the Rib-Deck Slab: The Construction

No propping system has been considered during the construction, hence, the only resisting element in this stage is the steel decking. The same loads considered for the beam" verification have been used here. Four LSs have been checked, (i) the ULS of the sagging bending moment in the edge span, (ii) the ULS of the hogging bending moment in the second support, (iii) the ULS of shear in the second support (for this case, the crippling of the steel decking webs has been considered to be the most restrictive shear failure mode) and, (iv) the SLS of deflection in the edge span during the concreting process limited to $d_2/180$ to control the ponding effect. In each case, the construction live load was assumed to be present in the worst spans. The resisting parameters of the steel decking have been taken from the Cofraplus 60 technical sheets.

### 3.2.2. Static LSs in the Rib-Deck Slab: The Service Life

During the service life, the composite section of the rib-deck slab must be verified. The same four LSs as for the construction stage have been checked but considering the resisting parameters of the composite section: (i) the ULS of sagging bending moment. For computing the resisting moment no reinforcement has been used, and the contribution of the steel decking has been accounted through using a degree of shear connection with the concrete (as done before with the beam" shear connection). The limit shear tension within the steel–concrete interface has been assumed to be 0.1 mPa [44]. (ii) The ULS of hogging bending moment. For computing the resisting moment, the contribution of the steel decking has been neglected, thus, a regular reinforced concrete section has been analyzed. An appropriate hogging reinforcement has been sized for each case being extended over a length equal to $d_2/3$ from the supports. To obtain both the acting sagging and hogging bending moments, a redistribution due to cracking of the bending law of a 15% has been assumed. (iii) The ULS of shear. The resisting shear of the composite slab

has been obtained as a result of two contributions, the shear resistance of the steel decking and the shear resistance of the concrete ribs without any shear reinforcement. (iv) The DSLS. The deflection at the edge span under the action of the worst live load placing has been computed as the average value between those obtained with and without considering cracking in the concrete. This has been limited to $d_2/350$.

## 4. VSLS of Floors Partially Treated with CLD

The VSLS of the floor has been assessed according to the simplified methodology provided by the AISC Steel Design Guide 11 [7]. This guideline has been reported to be the most accurate one when predicting the VSLS of composite floors according to [8,45]. The AISC method quantifies the vibration in terms of peak acceleration; however, this paper provides equivalent root mean square (rms) acceleration values, as these enable obtaining the response factors widely used in other guidelines and in future versions of Eurocodes to establish vibration limits.

*4.1. Modal Parameters of the Floor*

4.1.1. Fundamental Natural Frequency of the Floor

The fundamental natural frequency of a floor bay, $f_n$, has been obtained by computing the fundamental natural frequencies of the primary and secondary beams from the maximum static deflections, $\delta_1$ and $\delta_2$, under the floor's weight considered for the VSLS, and then applying the Dunkerle"s approximation as follows:

$$f_n = \frac{18}{\sqrt{1000\,(\delta_1 + \delta_2)}}.$$ (6)

These deflections have been computed considering a load per unit of length for each beam, $q_{VSLS,i}$, equal to:

$$q_{VSLS,i} = q_{SW,i} + q_{DL,i} + 0.1\,q_{LL,i},$$ (7)

where $q_{SW,i}$, $q_{DL,i}$ and $q_{LL,i}$ are the self-weight, dead load, and live load per unit of length of the beam, respectively. Then, the following expression has been derived to compute the deflection of a partially CLD-treated beam. The authors propose to apply the Maxwell-Mohr method to the law of elastic curvatures of a simply supported beam, with two different bending stiffnesses and uniformly loaded:

$$\delta_i = \frac{L_i^4\,q_{VSLS,i}}{384}\left[\frac{-3\lambda_i^4 + 8\lambda_i^3}{E_{st}\,I_{st,i}} + \frac{3\lambda_i^4 - 8\lambda_i^3 + 5}{E_c\,I_{c,i}}\right],$$ (8)

where $E_{st}$ and $E_c$ are the Young modulus of the steel and the concrete, respectively; $I_{st,i}$ is the moment of inertia of the steel profile used for the beams, and $I_{c,i}$ is a concrete-homogenized moment of inertia of the composite section present along the shear-connected region of the beam. $\lambda_i$ is the proportion of CLD-treated beam in parts per unit computed in Equation (1). For $1^{ry}$ beams just supporting one $2^{ry}$ beam at mid-span, $\delta_1$ has been increased by a factor of 1.3.

4.1.2. Effective Weight of the Floo's Fundamental Mode of Vibration

This magnitude has been computed following the AISC Design Guideline 11 as follows. This method is composed of 3 steps:

1. Computing the effective widths, $B_{eff,1}$ and $B_{eff,2}$, respectively associated with the simply supported bending modes of the 1$^{\text{ry}}$ and 2$^{\text{ry}}$ beams of the floor, following an already calibrated formulation described below:

$$B_{eff,1} = L_1 \left[ C_1 \left( \frac{I_{eff,2}/d_2}{I_{eff,1}/L_2} \right)^{1/4} \right] < L, \tag{9}$$

$$B_{eff,2} = L_2 \left[ C_2 \left( \frac{I_{slab}}{I_{eff,2}/d_2} \right)^{1/4} \right] < B, \tag{10}$$

where $I_{slab}$ is the concrete-homogenized moment of inertia of one meter of composite rib-deck slab, $C_1$ and $C_2$ are calibration coefficients depending on the type of composite floor beams and floor bay to be analyzed, and $I_{eff,1}$ and $I_{eff,2}$ are the effective concrete-homogenized moments of inertia of the 1$^{\text{ry}}$ and 2$^{\text{ry}}$ beams, respectively. They have been computed as follows:

$$I_{eff,i} = \frac{L_i^4 \, 5 \, q_{VSLS,i}}{384 \, E_c \, \delta_i}. \tag{11}$$

2. Computing the effective weights, $W_{eff,1}$ and $W_{eff,2}$, respectively associated with the simply supported bending modes of the 1$^{\text{ry}}$ and 2$^{\text{ry}}$ beams:

$$W_{eff,1} = \frac{B_{eff,1} \, L_1 \, q_{VSLS,1}}{2 \, L_2}, \tag{12}$$

$$W_{eff,2} = K_2 \, \frac{B_{eff,2} \, L_2 \, q_{VSLS,2}}{2 \, d_2}, \tag{13}$$

where $K_2$ is a factor equal to 1.5 for floor bays with adjacent bays having secondary beams of length higher than $0.7 \, L_2$, and equal to 1 in other cases.

3. Computing the final effective weight $W_{eff}$ of the combined mode of vibration:

$$W_{eff} = W_{eff,1} \, \frac{\delta_1}{\delta_1 + \delta_2} + W_{eff,2} \, \frac{\delta_2}{\delta_1 + \delta_2}. \tag{14}$$

4.1.3. Intrinsic Damping Ratio of the Floo's Fundamental Mode of Vibration

Two main sources of damping contribute to defining the final value of the fundamental modal damping ratio, $\xi_n$. These are: (i) the intrinsic damping of the structure, $\xi_{int}$ (the usual source of energy dissipation that depends on the type of floor structure, the floor finishes, and the furniture present on the floor) and, (ii) the additional damping ratio provided by the VE CLD treatment, $\xi_{CLD}$. Thus, the following equation has been used:

$$\xi_n = \xi_{int} + \xi_{CLD}. \tag{15}$$

$\xi_{int}$ has been computed assuming an electronic office fit-out with ceiling and ductwork. Hence, according to the AISC guideline, $\xi_{int} = 0.01 + 0.01 + 0.005 = 0.025$.

4.1.4. Additional Damping Ratio Provided by the CLD Treatment

In this paper, $\xi_{CLD}$ has been computed using a simplified new methodology proposed by the author based on solving the problem of a simply supported 'sandwich beam' partially treated with a VE core. This method is divided into 4 sub-steps.

1. Obtain $N_1$ and $N_2$, the number of 1$^{\text{ry}}$ and 2$^{\text{ry}}$ beams involved in the fundamental mode of vibration, respectively. This is performed using the effective widths, $B_{eff,i}$

computed before, assuming that they define the effective floor area contributing to the vibration when assessing the VSLS:

$$N_1 = \left( \frac{B_{eff,1}}{L_2} + 1 \right) \left( \frac{B_{eff,2}}{L_1} \right), \tag{16}$$

$$N_2 = \left( \frac{B_{eff,2}}{d_2} + 1 \right) \left( \frac{B_{eff,1}}{L_2} \right). \tag{17}$$

2. Compute the additional damping ratio, $\xi_{CLD,i}$, provided by the applied CLD treatments to isolated $1^{ry}$ and $2^{ry}$ beams. This has been done by analyzing a simply supported beam partially treated with CLD and characterized by the following parameters $L_i$, $L_{v,i}$, $P_i$, $R_d$, $h_c$, $b_{eff,i}$ and with a CLD treatment defined by $h_{v,i}$ and $b_{v,i}$. The value of $\xi_{CLD,i}$ depends on four dimensionless parameters:

   - The loss factor of the VE material $\eta_v$, the higher $\eta_v$ the better. In this paper, this parameter has been assumed to be constant and equal to 1.
   - The percentage of the beam length treated with CLD or %*CLD*. The higher this parameter, the higher the damping enhancement.
   - The so-called 'geometric parameter' $Y$, computed as the ratio between the following two bending stiffness:

$$Y = \frac{(E_c I_{c,i})}{(E_{st} I_{st,i})}. \tag{18}$$

   where the numerator and denominator are the bending stiffnesses of a 100% *CLD* beam where the VE core has been replaced by a shear connection with infinite or zero shear stiffness, respectively. This ratio oscillates between 1 and 3 for composite floor beams. The higher the $Y$, the higher the damping.

   - The so-called 'shear parameter' of the beam, $g$, which represents the shear stiffness of the VE core (it should be noted that $g$ is strongly dependent on the parameters of the VE treatment). This parameter is computed as follows:

$$g =''_v \frac{b_{v,i} \, L_i^2}{h_{v,i}} \frac{(E_{st} \, A_{st,i} + E_c \, A_{slab,i})}{E_{st} \, A_{st,i} \, E_c \, A_{slab,i}}, \tag{19}$$

   where $A_{st,i}$ and $A_{slab,i}$ are the areas of the profile and the slab that belong to the beam section to be analyzed, respectively. For each beam, there is an optimum $g_{opt}$ that provides the maximum additional damping to the beam. The value of $g_{opt}$ depends on $Y$, $\eta_v$ and %*CLD* of the beam as demonstrated in [46]. The dependency of $g_{opt}$ with $Y$ can be neglected in the range of $Y$ values adopted by floor beams. In the same way, the VE materials used for this purpose have similar values of $\eta_v$ close to 1, so this dependency can be also neglected.

Knowing this, the authors have proposed a simplified methodology to obtain $\xi_{CLD,i}$, assuming that it depends only on $g$ and %*CLD*. $\xi_{CLD,i}$ has been obtained using the set of curves depicted in Figure 7.

These curves have been obtained for a set of 15 m composite beams with a Cofraplus 60 rib-deck slab with a width of 3 m and a height of 0.12 m, made of lightweight concrete. The steel profiles used varied depending on the %*CLD* analyzed (UB 533X210X82 for 10 %*CLD*, UB 610X178X82 for 2—40 %*CLD*, UB 610X178X100 for 5—60 %*CLD*, and UB 610X210X133 for 7—100 %*CLD*). A detailed FE model of the beams was built using SHELL181 elements for the steel profile and SOLID185 elements for the VE treatment and the concrete slab. The applied CLD treatment had a thickness of 1 mm and a width equal to the width of the upper steel flange. The value of $\eta_v$ used was 1. Different complex modal analyses of each beam were performed varying the shear stiffness of the VE core to obtain the set of $\xi_{CLD,i}$ curves as a function of $g$.

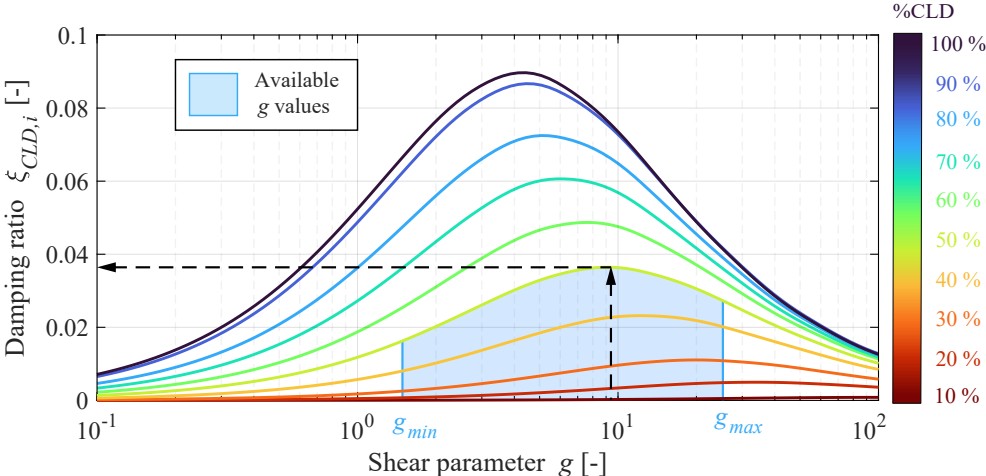

**Figure 7.** Set of design curves used to compute $\xi_{CLD,i}$ in a simply supported isolated beam of the floor.

Knowing this and given a partially treated floor beam, the geometry of the CLD treatment is designed by selecting the right values of $h_{v,i}$ and $b_{v,i}$ within the available ones, to provide a shear parameter as close as possible to $g_{opt}$, as shown in Figure 7 for a 50 %*CLD* beam.

3      Compute the bending modal strain energies, $U_{bend,i}$ of isolated $1^{ry}$ and $2^{ry}$ beams, as follows:

$$U_{bend,i} = \frac{1}{2} \int_0^{L_i} \left( \frac{M_{VSLS,i}(s)^2}{E(s)I(s)} \right) ds, \tag{20}$$

where $M_{VSLS,i}(x)$, $E(x)$ and $I(x)$ are the laws of bending moments (under the loads considered for the VSLS), Young modulus, and Moment of inertia along the beam length. Assuming the scheme of a simply supported beam uniformly loaded and divided into two regions of different inertia, the following expression is derived:

$$U_{bend,i} = \frac{q_{VSLS,i}^2 \, L_i^5}{1920} \left( \frac{3\,\lambda_i^5 - 15\,\lambda_i^4 + 20\,\lambda_i^3}{E_{st}I_{st,i}} - \frac{3\,\lambda_i^5 - 15\,\lambda_i^4 + 20\,\lambda_i^3 - 8}{E_c I_{c,i}} \right). \tag{21}$$

4      Calculate the final $\xi_{CLD}$ of the floor's fundamental mode of vibration as a weighted value of $\xi_{CLD,1}$ and $\xi_{CLD,2}$ depending on their contribution to the total modal strain energy of the floor:

$$\xi_{CLD} = \frac{\xi_{CLD,1} \, N_1 \, U_{bend,1} + \xi_{CLD,2} \, N_2 \, U_{bend,2}}{N_1 \, U_eend,1} + N_2 \, U_{bend,2}. \tag{22}$$

*4.2. Response of the Floor*

For checking the VSLS, two types of dynamic floor responses have been computed under walking excitation according to the methodology given by [7]: (i) the resonant response of vibration modes with frequencies lower than 9 Hz, and (ii) the impulsive transient produced by a footfall excitation.

4.2.1. Low-Frequency Resonant Response

A harmonic punctual load given by the following expression has been used to model the walking excitation acting at the anti-node of the vibration mode:

$$F_h(t) = Q \, \alpha_h \, sin(2 \, \pi \, f_n \, t), \tag{23}$$

where $t$ represents the time, $Q$ is the weight of an average human assumed to be 700 N and $\alpha_h$ is a dynamic loading factor depending on the loading frequency (assumed equal to $f_n$) and given by:

$$\alpha_h = 0.83\, e^{-0.35\, f_n}. \tag{24}$$

The resonant rms acceleration of the floor is then computed according to:

$$a_{rms,res} = \frac{K_{cres}\, Q\, 0.83\, e^{-0.35\, f_n}}{\sqrt{2}\, 2\, \xi_n \left( W_{eff}/a_g \right)}, \tag{25}$$

where $K_{res}$ is a factor equal to 0.5 that accounts for incomplete resonant build-up from walking, and, it considers the fact that the walker and the potentially annoyed person are not simultaneously at the same location of maximum modal acceleration. Finally, a resonant response factor, $R_{res}$ has been computed as follows (with $a_{rms,res}$ in m/s$^2$):

$$R_{res} = \frac{a_{rms,res}}{0.005}. \tag{26}$$

4.2.2. Impulsive Response of the Floor

The consecutive impulsive transient responses produced by a floor when excited by human footfalls may also be excessive in comfort terms; thus, this vibration has also been checked. For this the effective impulse function, $I_{eff}$ (with units of [Ns], derived by Willford and Young [6]) has been used as excitation:

$$I_{eff} = 42 \left( \frac{f_{step}^{1.43}}{f_n^{1.30}} \right), \tag{27}$$

where $f_{step}$ is the human pacing frequency. The use of higher pacing frequencies results in higher dynamic impulsive responses, thus, a $f_{step}$ = 2.6 Hz has been selected, as a conservative value. The initial peak acceleration just after the impulse is given by:

$$a_{p,imp} = \frac{K_c\, 2\pi f_n I_{eff}}{\left( W_{eff}/a_g \right)}, \tag{28}$$

where $K_c$ is a factor introduced by the AISC guideline equal to 1.3. This code also amplifies the impulsive response with a higher mode factor equal to 2, which considers the contribution of higher modes to the impulsive response. However, to the authors' knowledge, this factor is over-conservative and, in fact, it is not used in other guidelines as [6]. From here, $a_{rms,imp}$, the rms acceleration of the decaying response during the time lapse from one footfall to the next one, is computed according to:

$$a_{rms,imp} = \sqrt{ f_{step} \int_0^{(1/f_{step})} \left( a_{p,imp}\, e^{-2\pi f_n \xi_n t}\, sin(2\pi f_n t) \right)^2 dt}, \tag{29}$$

$$a_{rms,imp} \approx \frac{a_{p,imp}}{\sqrt{2}} \sqrt{ \frac{\left( 1 - e^{\frac{-4\pi f_n \xi_n}{f_{step}}} \right)}{\frac{4\pi f_n \xi_n}{f_{step}}} } \quad for \quad \xi_n < 0.1. \tag{30}$$

Finally, an impulsive response factor, $R_{imp}$ is computed as indicated below (with $a_{rms,imp}$ in m/s$^2$):

$$R_{imp} = \frac{a_{rms,imp}}{0.005} \ (f_n < 8 \text{ Hz}),$$

$$R_{imp} = \frac{a_{rms,imp}}{2\pi f_n \, 0.0001} \ (f_n > 8 \text{ Hz}).$$

$$\text{(31)}$$

### 4.2.3. Comparison with VSLS Limit

A limiting response factor, $R_{lim}$, must be set depending on the vibration comfort level required by the floor use. Hence, two possible safety factors for checking the VSLS can be built: (i) $SF_{Rmax,s}$ when considering checking both the resonant and impulsive responses, and (ii) $SF_{Rres,s}$ when only checking the resonant response of the floor:

$$SF_{Rmax,s} = \frac{max\left(R_{res}, R_{imp}\right)}{R_{lim}}, \tag{32}$$

$$SF_{Rres,s} = \frac{R_{res}}{R_{lim}}. \tag{33}$$

## 5. Optimization Problem Definition

In this section, the authors propose the definition of an optimization problem for the structural design of composite floors partially treated with CLD. The floor optimization addressed is a discrete multi-objective optimization problem with two OFs ($f_1(\underline{x})$ and $f_2(\underline{x})$) depending on a vector $\underline{x}$ of seven design variables defined by integer numbers (from $\underline{x}_1$ to $\underline{x}_7$), and with six design constraints defined by the functions $g_1(\underline{x})$ to $g_6(\underline{x})$. The optimization problem can be formulated as follows:

$$
\begin{aligned}
\min \ & f_1(\underline{x}) = M(\underline{x}) \\
\min \ & f_2(\underline{x}) = EC(\underline{x}) \\
\text{s.t.} \ & g_1(\underline{x}) = ULS_1(\underline{x}) \le 0 \\
& g_2(\underline{x}) = ULS_2(\underline{x}) \le 0 \\
& g_3(\underline{x}) = DSLS_1(\underline{x}) \le 0 \\
& g_4(\underline{x}) = DSLS_2(\underline{x}) \le 0 \\
& g_5(\underline{x}) = LS_{slab}(\underline{x}) \le 0 \\
& g_6(\underline{x}) = VSLS(\underline{x}) \le 0 \\
& 6 \le \underline{x}_1 = h_c \le 20 \\
& 1 \le \underline{x}_2 = Rd \le 4 \\
& 1 \le \underline{x}_3 = P_1 \le 94 \\
& 1 \le \underline{x}_4 = P_2 \le 94 \\
& 0 \le \underline{x}_5 = 10\,\lambda_1 \le 10 \\
& 0 \le \underline{x}_6 = 10\,\lambda_2 \le 10 \\
& 1 \le \underline{x}_7 = (N_2 - 1) \le 10 \\
& \underline{x} \in \mathbb{Z},
\end{aligned}
\tag{34}
$$

where $M(\underline{x})$ and $EC(\underline{x})$ are the functions to compute the mass of a floor bay and its embodied upfront carbon per unit of floor area, respectively. Additionally, $ULS_1(\underline{x})$, $ULS_2(\underline{x})$, $DSLS_1(\underline{x})$, $DSLS_2(\underline{x})$, $LS_{slab}(\underline{x})$ and $VSLS(\underline{x})$ are aggregate functions of the LSs checked for the different floor elements, and are described in detail in the following

subsection devoted to the constraints of the problem. They provide a value of 0 if the concerning LSs have been met and positive values if not.

*5.1. Design Variables*

The seven design variables are integers since the structural optimization of a floor is subjected to the adoption of discrete values on its geometrical parameters to ease the construction process of the floor.

The floor optimization performed assumes a given floor bay defined by fixed distances between columns $L_1$ and $L_2$ that belongs to a given floor with fixed total length ($L$) and fixed total width ($B$). This bay is optimized by finding the optimal values of the following variables: $h_c$, $R_d$, $P_1$, $P_2$, $10\lambda_1$, $10\lambda_2$ and ($N_2 - 1$). The remaining floor-defining parameters are obtained when checking the different LSs as follows:

- $n_i$: the number of shear studs used in each beam is calculated to meet the bending ULS of the beam at mid-span.
- $b_{v,i}$ and $h_{v,i}$: the dimensions of the CLD treatment used in each floor beam are obtained to maximize the $\xi_{CLD,i}$ of each beam.
- $\rho_{h,slab}$: the amount of hogging reinforcement of the slab is computed to meet the ULS of the hogging bending moment of the slab.

*5.2. Objective Functions*

Two objective functions have been minimized: (i) The mass of the floor bay per unit of area, $M(\underline{x})$, and (ii) the embodied upfront carbon of the floor bay per unit of area, $EC(\underline{x})$. The choice of these two OFs is based on the fact that to tackle the vibration problem from the design perspective, two main strategies can be adopted: (i) to increase the mass of the floor, by thickening the concrete slab, which has a great impact on the floor's $M(\underline{x})$ but not much in the $EC(\underline{x})$, or (ii) to stiffen the steel members of the floor, which increases much more the $EC(\underline{x})$ than the $M(\underline{x})$. This stays true when using low-strength concrete (with low cement content) and non-recycled steel, as in this case the carbon factor of the steel is much higher than the concrete one.

The computation of $EC(\underline{x})$ has been carried out according to the guideline 'How to Compute Embodied Carbon' by Gibbons et al. [47]. This study has only considered the influence of the embodied upfront carbon of the floor (i.e., from modules A1 to A5 of the floor's life-cycle). For the computation of $EC(\underline{x})$, the following floor elements have been considered: concrete slab, slab reinforcement, rib-deck sheet, steel beams, shear studs, and CLD treatment. Modules A1 to A3 (those belonging to the Product Stage of the life-cycle) are the main contributors to the final upfront EC of the floor. Thus, a list of the A1–A3 Carbon Factors used (denoted as $ECF_{A1-A3,n}$) for the $n$th materials of the floor, is provided in Table 4.

**Table 4.** Carbon Factors $ECF_{A1-A3,n}$ used for the different materials of the floor.

| Material | Floor Element | $ECF_{A1-A3,n}$ [kgCO$_2$eq/kg] |
|---|---|---|
| Concrete | Slab | 0.10 [1] |
| Steel | Reinforcement | 0.76 [1] |
| Steel | Beam profiles | 1.74 [1] |
| Steel | Studs | 1.74 [1] |
| Galvanized Steel | Rib-deck sheet | 2.87 [1] |
| Galvanized Steel | CLD Sheets | 2.87 [1] |
| VE material | CLD VE core | 6.00 [2] |

[1] Recommended default values for projects in the UK given by Table 2.3 of [47]. [2] Assuming the use of a VE material similar to the natural or butyl rubber. Taken from [48].

The floor studied is supposed to be built in a UK city. The steel and concrete have been assumed to be provided by a national and a local supplier, through road

transport of 300 km and 50 km, respectively. An 'A4 Carbon Factor' for road transport of 0.10749 gCO$_2$eq/kg/km has been used.

*5.3. Design Constrains*

Six different design constraint functions have been used: $ULS_1(\underline{x})$, $ULS_2(\underline{x})$, $DSLS_1(\underline{x})$ and $DSLS_2(\underline{x})$ for the ULS and DSLS of $1^{\text{ry}}$ and $2^{\text{ry}}$ beams, respectively; $LS_{slab}(\underline{x})$ for all the LSs of the rib-deck slab, and $VSLS(\underline{x})$ for the VSLS of the whole floor. These functions are built using safety factor functions $SF(\underline{x})$ (that compute for a given floor configuration $\underline{x}$, all the $SF$ listed in previous sections) in combination with the following Modified Heaviside function $H(SF)$ (that gives a value of 0 if the $SF$ value is lower than 1, and provides 1 if the input is greater or equal than 1):

$$
\begin{aligned}
H : \mathbb{R} &\to \{0,1\} \\
SF &\to H(SF) \\
H(SF) = \begin{cases} 1 & if \quad SF \geq 1 \\ 0 & if \quad SF < 1 \end{cases} &.
\end{aligned}
\tag{35}
$$

Thus, the constraint equations used are defined as follows:

$$
\begin{aligned}
ULS_i(\underline{x}) = 1 - [H(SF_{M+,c,i}(\underline{x})) + H(SF_{V,c,i}(\underline{x})) + H(SF_{M+,s,i}(\underline{x})) + \\
+ H(SF_{M+B,s,i}(\underline{x})) + H(SF_{V,s,i}(\underline{x}))]/5,
\end{aligned}
\tag{36}
$$

$$
DSLS_i(\underline{x}) = 1 - H(SF_{\delta LL,s,i}(\underline{x})),
\tag{37}
$$

$$
\begin{aligned}
LS_{slab}(\underline{x}) = 1 - [H(SF_{M+,c}(\underline{x})) + H(SF_{M-,c}(\underline{x})) + H(SF_{V,c}(\underline{x})) + H(SF_{\delta SW,c}(\underline{x})) + \\
+ H(SF_{M+,s}(\underline{x})) + H(SF_{M-,s}(\underline{x})) + H(SF_{V,s}(\underline{x})) + H(SF_{\delta LL,s}(\underline{x})))]/8,
\end{aligned}
\tag{38}
$$

$$
VSLS(SF(\underline{x})) = \begin{cases} 1 - H(SF_{Rmax,s}(\underline{x})) \\ 1 - H(SF_{Rres,s}(\underline{x})) \end{cases}.
\tag{39}
$$

These functions provide a value greater than zero if not all the LSs involving them are fulfilled, and equal to zero if all the LSs are met. The function $VSLS(\underline{x})$ has two possible definitions depending on whether the impulsive acceleration wants to be limited.

*5.4. Optimization Algorithm*

The evolutionary optimization algorithm 'Non-Dominated Sorting Genetic Algorithm II', best known as NSGA-II, for constrained and discrete multi-objective optimization, has been used in this paper. The algorithm version used is available in the python library 'pymoo' and has been implemented according to [49]. A random sampling of integer values has been performed to generate the initial solutions. A constraint handling method based on the principle 'Feasibility First' has been used to avoid the definition of any penalty function. This methodology was proposed by [50] and uses a fitness function (applied to each solution) that depends on the current population. When tournament selection is applied to these fitness values, feasible solutions are always emphasized over infeasible ones. A simulated binary crossover based on [50] with a crossover index $\eta_c$ = 3, and a probability of crossover for each variable of $p_c$ = 0.5, has been used. A polynomial mutation has been employed with a mutation parameter $\eta_m$ = 3 and a probability of mutation $p_m$ = 0.5. The histograms used for crossover and mutation have been rounded to deal with integer variables. Finally, for each optimization performed 100 generations with a population of 100, have been used.

## 6. Parametric Study

The optimization problem proposed has been used to develop a parametric study in which many different floor bays are optimized with and without CLD. This has been built by combining four different parameters which are needed to optimize a given floor bay:

- $L_i$: The length of the $1^{ry}$ and $2^{ry}$ beams of the floor. The parametric study has been focused on analyzing square floor bays, where $L_1 = L_2$. $L_i$ has been varied from 4.5 m to 19.5 m each 1.5 m, analyzing a whole set of 11 possibilities for this parameter.
- $R_{lim}$: The limiting response factor of the floor bay to define the VSLS. Four different possibilities have been considered for $R_{lim}$. First, a $R_{lim} = \infty$, which corresponds to floor designs in which the VSLS has not been checked, here denoted as 'Statically designed floors'. $R_{lim} = 8$, which is the limitation used for regular electronic offices. $R_{lim} = 4$ is the limit used for quiet spaces like silent offices or libraries. Finally, $R_{lim} = 2$ would apply to hospital floors.
- $SF_{VSLS}$: The $SF$ used to define the VSLS. Two possibilities are contemplated, $SF_{Rmax}$ which limits the impulsive and resonant floor responses, and $SF_{Rres}$ which only limits the resonant response.
- $CLD_{int}$: This indicates if the CLD treatment has been integrated into the design. Two possibilities are studied, $CLD_{int} = $ CLD and $CLD_{int} = $ NO CLD.

The following vectors provide a summary of the four parameters used and the values they can adopt:

$$L_i = [4.5, \ 6.0, \ 7.5, \ 9.0, \ 10.5, \ 12.0, \ 13.5, \ 15.0, \ 16.5, \ 18.0, \ 19.5],$$

$$R_{lim} = [2, \ 4, \ 8, \ \infty],$$

$$SF_{VSLS} = [SF_{Rmax}, \ SF_{Rres}],$$

$$CLD_{int} = [NO \ CLD, \ CLD].$$

(40)

A total of 176 cases (resulting from combining all the different possibilities $11 \times 4 \times 2 \times 2 = 176$ ) have been analyzed. One particular case can be, for example, the one defined by the vector [9, 4, $SF_{Rmax}$, NO CLD], which corresponds to a 9 m $\times$ 9 m floor bay, designed to comply with a maximum response factor of 4, including the impulsive and resonant responses, and without any integrated CLD treatment.

For each optimized floor bay seven design parameters result from the optimization problem if the CLD is integrated: $h_c$, $R_d$, $P_1$, $P_2$, $\lambda_1$, $\lambda_2$ and $(N_2 - 1)$. If the CLD is not used, this number reduces to five as $\lambda_1$ and $\lambda_2$ are set to 0.

The geometry configuration of the whole floor to which the floor bays optimized belong is a row of bays sharing a secondary beam between them, as depicted in Figure 8.

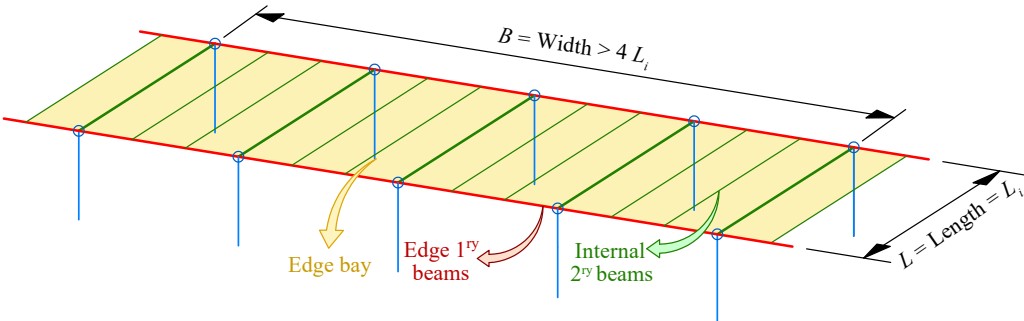

**Figure 8.** Floor configuration composed of edge bays used for the parametric study.

### 6.1. Results

For each case, a multi-objective optimization has been performed. Figures 9 and 10 provide the Pareto fronts of the achieved global optima for different floor spans, different

vibration limitations and, with and without CLD treatment. The first one has been obtained using $SF_{Rmax}$ and the second one making use of $SF_{Rres}$. Shaded areas in green, blue, and red represent the difference in the Pareto space between the optimal designs obtained without and with CLD. The bigger these areas, the higher the effectiveness of the CLD treatment.

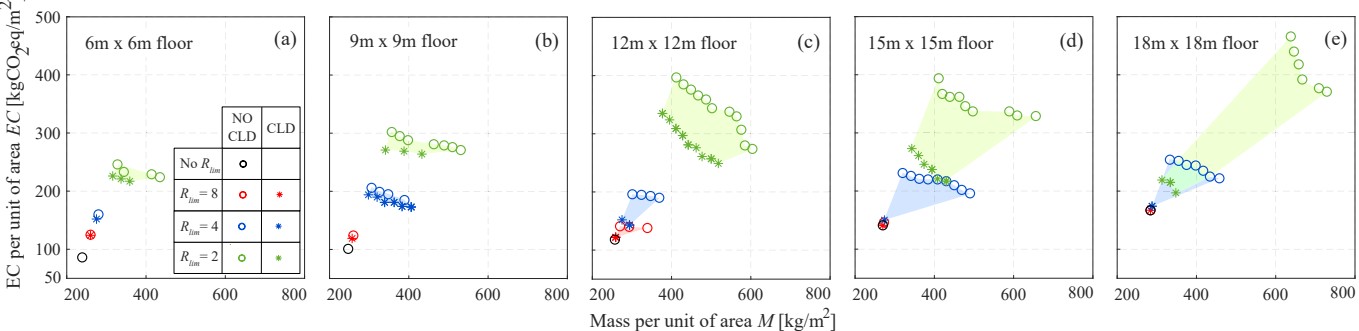

**Figure 9.** Pareto fronts of the floors in terms of $M$ and $EC$ designed with $SF_{Rmax}$ and for different spans: (**a**) 6 m. (**b**) 9 m. (**c**) 12 m. (**d**) 15 m (**e**) 18 m.

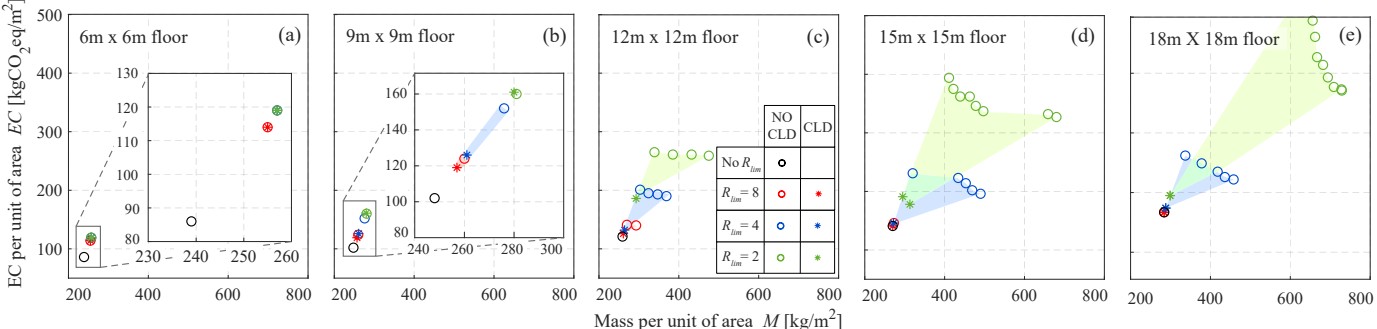

**Figure 10.** Pareto fronts of the floors in terms of $M$ and $EC$ designed with $SF_{Rres}$ and for different spans: (**a**) 6 m. (**b**) 9 m. (**c**) 12 m. (**d**) 15m (**e**) 18 m.

For each studied case, the lightest designs (those with an optimal mass $M$ and located at the left edge of each Pareto front) have been represented in terms of their $M$ and $EC$ depending on the span. Figures 11 and 12 correspond to floors designed using $SF_{Rmax}$ and $SF_{Rres}$, respectively. Polynomial trend curves have been included in these charts for a better interpretation of the results. The shaded areas in these figures have the same function as in the previous ones.

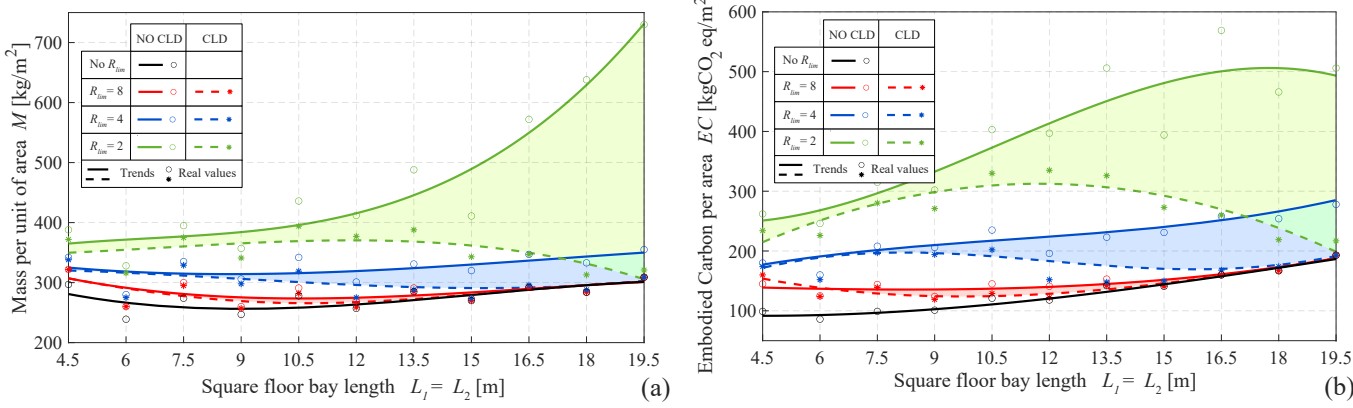

**Figure 11.** Floors designed with $SF_{Rmax}$, optimal designs in terms of mass per square meter. (**a**) Mass per square meter $M$. (**b**) Embodied carbon per square meter $EC$.

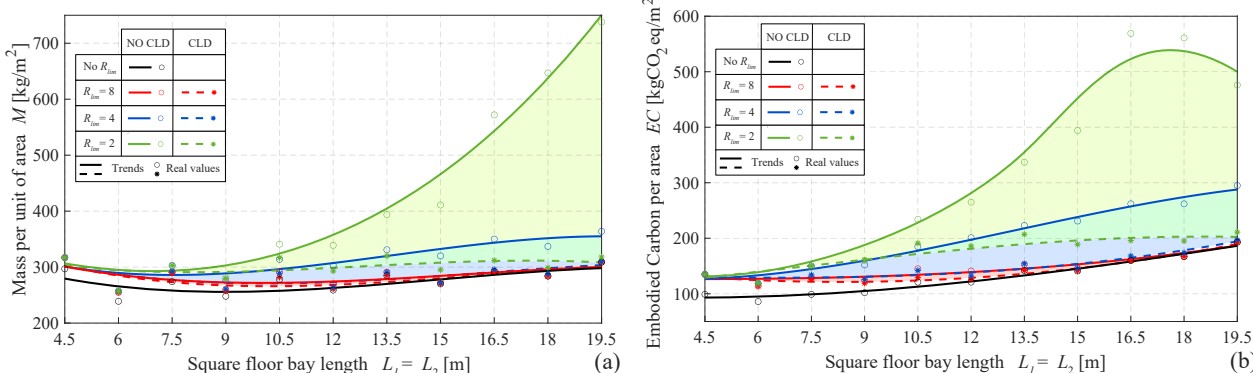

**Figure 12.** Floors designed with $SF_{Rres}$, optimal designs in terms of mass per square meter. (**a**) Mass per square meter $M$. (**b**) Embodied carbon per square meter $EC$.

Finally, within the previous figures, the black circles and lines represent the static designs in which the VSLS has not been checked. When imposing a more restrictive vibration limitation, these floor designs must be changed in terms of mass, stiffness, or damping to meet the new VSLS. To provide a meaningful interpretation of these, the $R_{res}$ and the $R_{imp}$ of these statically designed floors have been represented together with their $f_n$ and $W_{eff}/g$ in Figure 13a,b, respectively. Shaded areas in Figure 13a represent the vibration limitation used in the parametric study. Shaded areas in Figure 13b represent the frequency restriction of 3 or 4 Hz that some design codes impose on long-span floors to avoid excessive low-frequency vibrations.

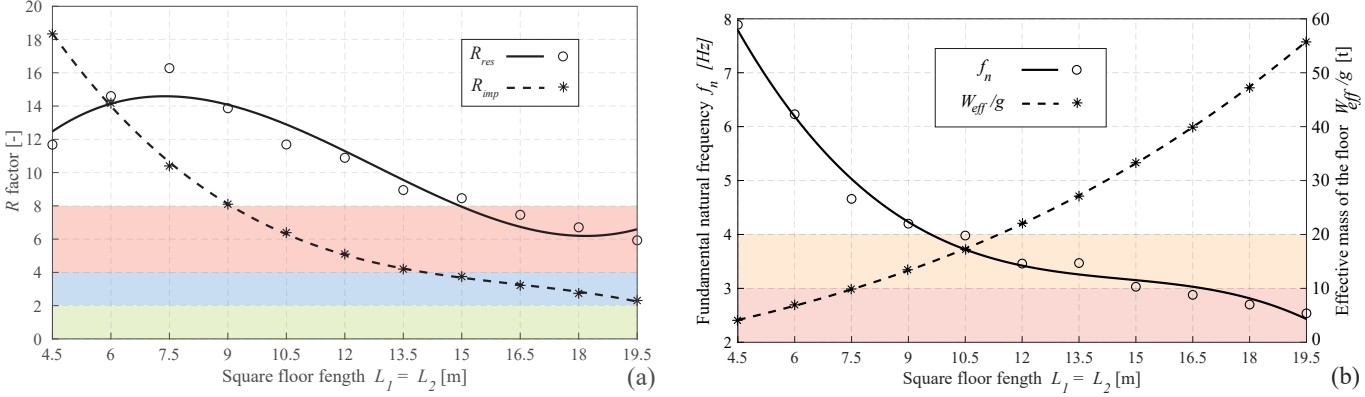

**Figure 13.** Statically designed floors without CLD and with $R_{lim} = \infty$. (**a**) Dynamic response of the floors in terms of $R_{res}$ and $R_{imp}$. (**b**) Dynamic parameters of the floors $f_n$ and $W_{eff}$.

### 6.2. Discussion

The first finding to note in Figures 9–12 is that the statically designed floors are the ones with less $M$ and $EC$. Also, the Pareto front in these cases converges to one point, as the concrete slab is kept as light as possible.

It is also clear from the results obtained that the more restrictive the vibration limitation, the higher the oversizing of the floor to meet the VSLS.

In floors designed without CLD, when imposing more restrictive $R_{lim}$ values (especially 4 and 2), the Pareto front begins to open. This widening effect is more evident in two cases:

- In long-span floors with span > 12 m (as can be seen in Figures 9d,e and 10d,e). This can be explained by observing in Figure 13a,b that long-span floors have a dynamic response dominated by $R_{res}$. They also have low values of $f_n$. On the one hand, the excessive resonant vibration may be tackled by stiffening the steel profiles of the floor (which implies a high $EC$ cost, but has a low repercussion in $M$) to increase its $f_n$ and

thus, reduce the amplitude of the exciting human harmonic $\alpha_h$. On the other hand, a solution with less $EC$ cost but more impact on the final floor's $M$, is to thicken the concrete slab to reduce the floor resonant response. These two possibilities are the extreme solutions of the Pareto fronts.

- In short-span floors designed using $SF_{Rmax}$ (compare, for example, Figure 9a,b with respect to Figure 10a,b). Again, looking at Figure 13a,b, it is noticeable that short-span floors have higher values of $R_{res}$ and $R_{imp}$, higher values of $f_n$, and lower values of $W_{eff}$. In these floors, $R_{res}$ can be optimally reduced with a minimum stiffening of the floor that effectively rises $f_n$ the right amount to meet the VSLS (this explains why in Figure 10a,b, the Pareto front converges to 1 point). Nevertheless, reducing the floors' $R_{imp}$ requires, either a major stiffening of the floor, (which reduces the value of the effective impulse loading $I_{eff}$) or a substantial increase of its $W_{eff}$ (to reduce the impulsive response). Hence, controlling the impulsive vibration $R_{imp}$ produces a significant oversizing of short-span floors compared to when it is not controlled (see results of Figures 11 and 12 for span values lower than 12 m).

Regarding the effectiveness of the CLD integration in the final design of the floor, the following conclusions may be extracted:

- The CLD integration enables increasing the damping ratio of the floor $\xi_n$. Hence, it is mainly effective when implemented in floors in which the resonant response $R_{res}$ is the dominant one, i.e., long-span floors with values of $R_{lim}$ around 4 or 2. This can be appreciated in Figures 9d,e and 10d,e, and also a bit in Figures 9c and 10c. Table 5 provides 15 m and 18 m floors, the percentage of oversizing with respect to the statically designed cases (those not meeting the VSLS), for solutions located at the middle of the Pareto front. When CLD is not used, this oversizing in terms of $M$ and $EC$ is around 100% for floors with $R_{lim}$ = 2, and around 50% for floors with $R_{lim}$ = 4. When the CLD is used the oversizing decreases to around 20% for the $M$, and an average value of 34% for the $EC$, in floors with $R_{lim}$ = 2. When $R_{lim}$ = 4 the oversizing decreases to 2% for the $M$, and 5% for the $EC$. This decrease is even more evident when checking the VSLS using $SF_{Rres}$.
- The CLD does not result as effective in short-span floors (those with a span < 10 m). On the one hand, When assessing the VSLS according to $SF_{Rmax}$, a substantial big increase of $\xi_n$ does not have a significant impact on reducing the impulsive response of the floor. On the other hand, when using $SF_{Rres}$, the resonant response of these floors seems to be better tackled by minimally stiffening the floors rather than increasing their damping. Both conclusions are clear when looking at Figures 11 and 12 for spans lower than 10 m.
- Floors with a span of around 12 m represent a transition zone between short-span and long-span floors. In this intermediate range of spans the CLD efficacy is perceptible but not as high as on long-span floors.
- For floor designs in which $M$ is the minimum possible, (as those depicted in Figures 11 and 12) the CLD effect can be summarized as follows: for floors with a $R_{lim}$ = 4, the CLD enables a reduction by around 50 kg/m$^2$ and 100 kgCO$_2$eq/m$^2$ with respect to the cases when it is not used. In floors with $R_{lim}$ = 2, this reduction increases with the span, with an average of 200 kg/m$^2$ and 250 kgCO$_2$eq/m$^2$.

**Table 5.** Oversizing in terms of $M$ and $EC$ of floor designs with 15 m and 18 m span, that meet different VSLSs, with respect to the statically designed floors ($R_{lim} = \infty$).

| VSLS Type | $L_i[m]$ | CLD Treatment | $M$ Oversizing [%] | $EC$ Oversizing [%] |
|---|---|---|---|---|
| $SF_{Rmax} - R_{lim} = 2$ | 15 | NO CLD | 84 | 139 |
| | | CLD | 45 | 70 |
| | 18 | NO CLD | 134 | 134 |
| | | CLD | 18 | 28 |

**Table 5.** *Cont.*

| VSLS Type | $L_i[m]$ | CLD Treatment | *M* Oversizing [%] | *EC* Oversizing [%] |
|---|---|---|---|---|
| $SF_{Rres} - R_{lim} = 2$ | 15 | NO CLD | 84 | 139 |
| | | CLD | 9 | 35 |
| | 18 | NO CLD | 134 | 155 |
| | | CLD | 5 | 16 |
| $SF_{Rmax} - R_{lim} = 4$ | 15 | NO CLD | 51 | 56 |
| | | CLD | 2 | 7 |
| | 18 | NO CLD | 40 | 46 |
| | | CLD | 2 | 5 |
| $SF_{Rres} - R_{lim} = 4$ | 15 | NO CLD | 60 | 58 |
| | | CLD | 1 | 4 |
| | 18 | NO CLD | 40 | 43 |
| | | CLD | 2 | 4 |

## 7. Conclusions

This paper studies the integration of CLD treatments into the design workflow of composite floors prone to vibrate. This treatment consists of a thin VE layer included between the steel member and the concrete slab of composite floor beams for a proportion of their length near the supports. This technology enables increasing the floor-damping ratio when vibrating in vertical bending modes, which allows for the reduction in the amount of additional mass or stiffness typically increased to overcome the VSLS.

A constrained discrete multi-objective optimization problem has been proposed to design a floor bay with different CLD treatments applied on their 1[ry] and 2[ry] beams. Seven design variables of the floor have been considered The design constraints of the problem are the different LSs of the floor, and two OFs have been used: the embodied carbon and the mass of the floor per unit of area.

Finally, a parametric study for different square floor bays, with spans varying from 4.5 to 19.5 m, has been carried out to compare the optimal structural solutions obtained with and without making use of the CLD treatment. Four different limits of vibration have been imposed from less to more restrictive. The results obtained indicate that for long-span floors (>12 m) the reduction in terms of mass and EC is substantial. The CLD enables the reduction in the structural oversizing in terms of mass from values of 100% or 50% to around 20% or 2% for floors meeting a VSLS limited by *R* factors of 2 or 4, respectively. Moreover, when the impulsive vibration of the floor is not checked, this enhancement is even higher.

Future work will be focused on studying the efficacy of this CLD treatment when implemented on lightweight concrete and timber floors.

**Author Contributions:** Conceptualization, C.M.C.R., I.M.D. and C.G.-C.; methodology, C.M.C.R., I.M.D., J.H.G.-P. and C.G.-C.; software, C.M.C.R., J.H.G.-P. and C.G.-C.; validation, C.M.C.R. and C.G.-C.; formal analysis, C.M.C.R.; investigation, C.M.C.R.; resources, C.M.C.R. and C.G.-C.; data curation, C.M.C.R.; writing-original draft preparation, C.M.C.R.; writing-review and editing, C.M.C.R., I.M.D., J.H.G.-P. and C.G.-C.; visualization, C.M.C.R.; supervision, I.M.D. and J.H.G.-P.; project administration, I.M.D. and J.H.G.-P.; funding acquisition, I.M.D. and J.H.G.-P. All authors have read and agreed to the published version of the manuscript.

**Funding:** The authors acknowledge Grant PID2021-127627OB-I00 (Transport Infrastructures subjected to dynamic loading: assessment techniques for the sustainability, intelligent maintenance and comfort) funded by Ministerio de Ciencia e Innovación, Agencia Estatal de Investigación and 10.13039/501100011033 FEDER, European Union. Christian Gallegos-Calderón expresses his gratitude to the Secretariat of Higher Education, Science, Technology and Innovation of Ecuador (SENESCYT) for the PhD scholarship CZ02-000167-2018.

**Acknowledgments:** The authors would like to thank the help provided by Pablo Vidal Fernández on the previous developments related to this work.

**Conflicts of Interest:** The authors declare no conflict of interest.

## Abbreviations

| | |
|---|---|
| AISC | American Institute of Steel Construction |
| CLD | Constrained layer damping |
| DSLS | Deflection serviceability limit state |
| DL | Dead load |
| EC | Embodied carbon |
| HFF | High-frequency floors |
| HIP | Heathcote Industrial Polymers |
| LL | Live load |
| LFF | Low-frequency floors |
| LS | Limit state |
| NSGA-II | Non-Dominated Sorting Genetic Algorithm |
| OF | Objective function |
| SCI | Steel Construction Institute |
| SF | Safety factor |
| SW | Self-weight |
| ULS | Ultimate limit state |
| VE | Visco-elastic |
| VSLS | Vibration serviceability limit state |
| $1^{ry}$ | Primary |
| $2^{ry}$ | Secondary |
| $A_{st,i}$ | Area of the steel profile of beam '$i$' |
| $A_{slab,i}$ | Area of the slab section belonging to the section of the composite beam '$i$' |
| $a_g$ | Acceleration of gravity |
| $a_{p,imp}$ | Impulsive peak acceleration |
| $a_{rms,res}$ | Root mean square resonant acceleration |
| $a_{rms,imp}$ | Root mean square impulsive acceleration |
| $B$ | Total width of the floor parallel to the primary beams |
| $B_{eff,i}$ | Effective width of the area involved in the fundamental mode of vibration of beams '$i$' |
| $b_{eff,i}$ | Effective breadth of the slab of the composite beam '$i$' |
| $b_{f,i}$ | Width of the top steel flange of the beam '$i$' |
| $b_{v,i}$ | Width of the VE core of the CLD treatment applied to the beam '$i$' |
| $C_1$ | Calibration coefficient |
| $C_2$ | Calibration coefficient |
| $DSLS$ | Functions to compute the DSLS of primary and secondary floor beams |
| $d_2$ | Separation between secondary beams |
| $E_c$ | Young modulus of concrete |
| $E$ | Law of Young modulus along the beam '$i$' |
| $E_{st}$ | Young modulus of steel |
| $EC$ | Embodied carbon per unit of area of the floor |
| $F_h$ | Human-induced dynamic punctual force |
| $f_n$ | Fundamental natural frequency of the floor |
| $f_{step}$ | Pacing frequency of the human walking load |
| $f_u$ | Ultimate tensile strength |
| $f_y$ | Steel yielding stress |
| $f_1, f_2$ | Objective function |

| | |
|---|---|
| $G'_v$ | Storage modulus of the VE material |
| $g$ | Shear parameter |
| $g_{opt}$ | Optimum shear parameter |
| $g_1 - g_6$ | Design constraint functions |
| $H$ | Modified Heaviside function |
| $h_c$ | Height of the concrete slab over the ribs in cm |
| $h_{v,i}$ | Height of the VE core of the CLD treatment applied to the beam '$i$' |
| $I_{c,i}$ | Concrete-Homogenized Moment of the composite section of beam '$i$' |
| $I_{eff}$ | Effective impulse due to human footfall |
| $I_{eff,i}$ | Effective concrete-homogenized moment of inertia of beam '$i$' |
| $I_{slab}$ | Concrete-homogenized moment of inertia of 1 m of slab |
| $I_{st,i}$ | Moment of inertia of the steel profile of beam '$i$' |
| $i$ | Subscript to indicate the type of beam: 1 for primary and 2 for secondary |
| $K_c$ | Calibration factor |
| $K_{cre}$ | Resonant build-up factor |
| $K_2$ | Calibration factor |
| $L$ | Total length of the floor parallel to the secondary beams |
| $L_i$ | Length of the beam '$i$' |
| $L_{v,i}$ | Half of the CLD-treated length of a floor beam '$i$' |
| $LS_{slab}$ | Functions to compute the LSs of concrete slab |
| $M$ | Mass per unit of area of the floor |
| $M_{Ed}$ | Design sagging bending moment |
| $M_{Ed,s}$ | Service-life design sagging bending moment of a beam |
| $M_{p,st}$ | Plastic bending moment of a steel profile |
| $M_{Rd}$ | Resisting sagging bending moment |
| $M_{Rd,comp}$ | Resisting sagging bending moment of a composite steel–concrete section |
| $M_{Rd,st}$ | Resisting sagging bending moment of a steel section |
| $M_{VSLS,i}$ | Bending moments law under the loads considered for the VSLS in beam '$i$' |
| $Nc$ | Axial force in the concrete when a partial degree of shear connection is considered. |
| $Ncf$ | Axial force in the concrete when the full degree of shear connection is considered. |
| $N_i$ | Number of beams of type '$i$' involved in the fundamental vibration mode of the floor |
| $n$ | Total number of shear studs |
| $P_i$ | Profile number associated with the beam '$i$' |
| $P_{rd}$ | Shear resisting force of a shear stud |
| $p_c$ | Crossover probability |
| $p_m$ | Probability of mutation |
| $Q$ | Average human weight |
| $q_{DL,i}$ | Load per unit of length in beam '$i$' due to deal load |
| $q_{LL,i}$ | Load per unit of length in beam '$i$' due to live load |
| $q_{SW,i}$ | Load per unit of length in beam '$i$' due to self-Weight |
| $q_{VSLS,i}$ | Load per unit of length in beam '$i$' considered for the VSLS |
| $R_d$ | Number to designate the rib-deck |
| $R_{imp}$ | Impulsive response factor |
| $R_{res}$ | Resonant response factor |
| $SF_{M+,c}$ | Safety factor for sagging bending moment of the slab in construction |
| $SF_{M-,c}$ | Safety factor for hogging bending moment of the slab in construction |
| $SF_{M+,s}$ | Safety factor for sagging bending moment of the slab in service life |
| $SF_{M-,s}$ | Safety factor for hogging bending moment of the slab in service life |
| $SF_{M+,c,i}$ | Safety factor for sagging bending moment in the construction of beam '$i$' |
| $SF_{M+,s,i}$ | Safety factor for sagging bending moment at mid-span in the service life of beam '$i$' |
| $SF_{M+B,s,i}$ | Safety factor for sagging bending moment at section B in the service life of beam '$i$' |
| $SF_{Rmax,s}$ | Safety factor of VSLS considering impulsive and resonant floor response. |
| $SF_{Rres,s}$ | Safety factor of VSLS considering only resonant floor response. |

| $SF_{V,c}$ | Safety factor for shear of the slab in construction |
|---|---|
| $SF_{V,s}$ | Safety factor for shear of the slab in service life |
| $SF_{V,c,i}$ | Safety factor for shear in construction of beam '$i$' |
| $SF_{\delta LL,s}$ | Safety factor for deflection of the slab in service life |
| $SF_{\delta LL,s,i}$ | Safety factor for deflection at mid-span of beam '$i$' |
| $SF_{\delta SW,c}$ | Safety factor for deflection of the slab in construction |
| $s$ | Longitudinal spatial coordinate along a beam element |
| $t$ | Time variable |
| $U_{bend,i}$ | Modal strain energy of bending of a beam '$i$' |
| $ULS$ | Functions to compute the ULS of primary and secondary floor beams |
| $V_{p,st}$ | Plastic shear force of the web of a steel profile |
| $VSLS$ | Function to compute the VSLS of the floor |
| $W_{eff}$ | Effective weight associated with the fundamental mode of vibration of a floor bay |
| $W_{eff,i}$ | Effective weight associated with the fundamental mode of vibration of beam '$i$' |
| $\underline{x}$ | Vector of design variables |
| $Y$ | Geometric parameter of a CLD-treated beam |
| $\alpha_h$ | Dynamic loading factor |
| $\Delta\xi$ | Additional damping ratio |
| $\delta_i$ | Maximum static deflection of beam '$i$' under the loads considered for the VSLS |
| $\eta_c$ | Parameter of crossover |
| $eta_m$ | Parameter of mutation |
| $\eta_v$ | Loss factor or the VE material |
| $\lambda_i$ | Parts per unit of CLD-treated length in a beam with subscript '$i$' |
| $\xi_n$ | Modal damping ratio of the fundamental mode of vibration |
| $\xi_{int}$ | Damping ratio of the floor due to its intrinsic energy dissipation capacity |
| $\xi_{CLD}$ | Additional modal damping ratio of the floor provided by the CLD treatment |
| $\xi_{CLD,i}$ | Additional modal damping ratio provided by the CLD treatment to a beam '$i$' |
| $\rho$ | Density |

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
