# Peer review of "Structural Optimization of Lightweight Composite Floors with Integrated Constrained Layer Damping for Vibration Control"

_actuators, doi:10.3390/act12070288_

Round 1
Reviewer 1 Report
The article is very interesting and very comprehensive. The issues discussed are generally known in mechanical engineering, for example, the foundation of large machines, marine engines and many others. Also in construction, similar solutions are commonly used, disturbing the propagation of waves coming from communication excitations or constituting an element of passive or active vibroinsulation.
This does not detract from the effort put into the preparation of the article. However, I have a few questions, suggestions and reservations.
1. I have reviewed the topics of articles published in "Actuators" and I believe that the submitted article does not fit the journal's profile. I don't find any aspect related to controls or active techniques here, apart from mentioning active techniques in the introduction. Please consider whether this is the right place for this publication.
2. There are a lot of abbreviations in the article. Many times, while reading a rather long text, I had to look for the meaning of many of them in the earlier pages. I suggest adding an index of the most important abbreviations and symbols.
3. Many of the equations used are engineering (approximate). Understanding the limitations, and at the same time not being a constructor of buildings, I can accept it.
4. Figure 12 is lost.
Author Response
The authors acknowledge the review provided by Reviewer 1. The answer to the comments of reviewer 1 is available in the attached '.pdf' document. The authors have added both lists of acronyms and symbols at the end of the paper, according to the indications given by reviewer 1.

Reviewer 2 Report
1. The theoretical part is well written, However, the experiments are not explained in enough detail.
2. The composite floor should be described according to the properties of the material and composition.
3. The methodology described should refer to standards for measurement of damping characteristics.
4. In page 6, line187, it is mentioned "In this paper, the design of a composite floor with integrated CLD treatments will be done according to the floor bay depicted in Figure 4". The sentence should be modified. Why "will be" done? It should be "was" done.
5. The abstract is too brief whereas the conclusion section is too long. These sections need revising.
6. There is some error in Figure 12. It does not appear in the boxes.
7. Why is there a short section for results and another short section for discussion? They should be combined together.
8. A major part of the paper looks like literature review, being taken from other sources. It is not very clear what is done and what is reported from other literature.
9. It is suggested that the structure of the paper be changed to a standard division into sections e.g., Introduction, Methodology, Results & Discussion and Conclusions.
10. In the current form, it looks like a "book chapter" and not a "Research paper".
Quality of English Language is correct, but should be checking for grammar mistakes.
Author Response
The authors acknowledge the review provided by Reviewer 2. The answers to the comments given by the reviewer can be found in the attached '.pdf' document. Some changes have been made in the revised version due to the suggestions provided by Reviewer 2.
